# Build, Share and Remix: 3D Printing for Speeding Up the Innovation Cycles in Ambient Ionisation Mass Spectrometry (AIMS)

**DOI:** 10.3390/metabo12020185

**Published:** 2022-02-17

**Authors:** Nancy Shyrley García-Rojas, Héctor Guillén-Alonso, Sandra Martínez-Jarquín, Abigail Moreno-Pedraza, Leonardo D. Soto-Rodríguez, Robert Winkler

**Affiliations:** 1Department of Biotechnology and Biochemistry, Center for Research and Advanced Studies (CINVESTAV) Irapuato, Km. 9.6 Libramiento Norte Carr. Irapuato-León, Irapuato 36824, Mexico; shyrley.garcia@cinvestav.mx (N.S.G.-R.); hector.guillen@cinvestav.mx (H.G.-A.); abigail.moreno@cinvestav.mx (A.M.-P.); leonardo.soto@cinvestav.mx (L.D.S.-R.); 2Department of Biochemical Engineering, Nacional Technological Institute, Celaya 38010, Mexico; 3Department of Chemistry and Applied Biosciences, ETH Zürich, 8093 Zürich, Switzerland; sandra.martinez@org.chem.ethz.ch

**Keywords:** ambient ionisation, mass spectrometry, 3D printing

## Abstract

Ambient ionisation mass spectrometry (AIMS) enables studying biological systems in their native state and direct high-throughput analyses. The ionisation occurs in the physical conditions of the surrounding environment. Simple spray or plasma-based AIMS devices allow the desorption and ionisation of molecules from solid, liquid and gaseous samples. 3D printing helps to implement new ideas and concepts in AIMS quickly. Here, we present examples of 3D printed AIMS sources and devices for ion transfer and manipulation. Further, we show the use of 3D printer parts for building custom AIMS sampling robots and imaging systems. Using 3D printing technology allows upgrading existing mass spectrometers with relatively low cost and effort.

## 1. Introduction

Mass spectrometry (MS) is a central method in analytical chemistry because it can analyse complex mixtures of molecules with high sensitivity and selectivity. However, conventional MS techniques require an adequate sample workup, including extracting compounds from tissues by mixing with organic solvents, centrifugation, and filtration. Such procedures are not suitable for directly studying metabolites in their natural context.

The first ambient ionisation mass spectrometry (AIMS) methods were reported in 2004 and 2005. Desorption electrospray ionisation (DESI) [1] and electrosonic spray ionisation (ESSI) [2] are based on electrospray ionisation, using different principles for lifting the molecules into the gas phase under ambient temperature and pressure. Direct analysis in real-time (DART) operates a plasma beam for the desorption and ionisation of molecules from solid, liquid and gaseous samples [3].

Since then, a vast diversity of ambient ionisation methods, respective acronyms, and applications have been reported. However, generally, AIMS sources can be classified into either spray-, plasma-, or chemical ionization-based [4,5,6,7]. The solvent and gas flow techniques desorb molecules mainly by momentum transfer; plasma and chemical ion sources also employ thermal desorption. Combined AIMS methods often use a laser for desorption, with following post-ionization. In this case, the compounds are released by thermal desorption or energy-sudden activation, and the ionization mode is defined by the coupled ion source [4].

Despite their simplicity, AIMS methods can drastically expand the range of detectable molecules, e.g., for measuring highly hydrophobic compounds [8] and to detect semivolatile and volatile metabolites from biological tissues [9].

For the trace detection of volatile organic compounds (VOCs), several MS methods are well-established, for example, membrane-inlet (MI) MS [10], selected-ion flow-tube (SIFT) MS [11], and proton-transfer-reaction (PTR) MS [12]. AIMS techniques are not limited to compounds in gas phase, but combine desorption and ionisation processes, and are therefore suitable for solid, liquid and gaseous substances [4]. Terms such as ‘ambient desorption/ionisation’ or ‘ambient sampling/ionisation’ would be more precise, but the short ‘ambient ionisation’ MS has been adopted by the MS community for these methods. AIMS suffers drawbacks such as matrix effects. However, the possibility to quickly obtain metabolic profiles for native biological materials without prior sample work-up, makes AIMS very attractive for metabolomic studies [13].

Most AIMS sources have a simple technical design and provide new analytical options for existing MS infrastructures, such as high-throughput sampling [14,15] and imaging [16,17]. Yet, surprisingly, only a few of them are commercially available and at elevated costs.

Ambient ionisation methods are less demanding for materials than conventional techniques because the ionisation takes place outside the vacuum system of the mass spectrometer. Therefore, the 3D printing of AIMS ion sources and ion manipulation devices is possible. Moreover, 3D printing provides a cheap and fast way to manufacture diverse custom pieces and assemble experimental prototypes for ion mobility and mass spectrometry [18]. Besides, components and software of 3D printers serve for building custom AIMS robots [19].

Here, we present examples of using 3D printing for creating AIMS systems, highlighting its tremendous potential in analytical chemistry.

## 2. AIMS and 3D Printing
Technology

Grajewski et al. [20] provides an excellent review on 3D printing techniques and materials used for mass spectrometry applications [20], and our group published a mini-review about the emerging role of 3D printing in ion mobility spectrometry and mass spectrometry [18]. Therefore, we will focus on applications of 3D printing that are directly related to ambient ionisation mass spectrometry (AIMS).

Figure 1 gives an overview of 3D printed parts in AIMS. In Table 1, we summarise AIMS gadgets and applications.

We noted some basic concepts in the design of 3D printed AIMS components:

### 2.1. Part Design

In additive manufacturing, every slice is built upon the previous layer. Therefore, the 3D printing of parts with overhangs is limited with some technique, such as the widely used fused deposition modelling (FDM). In contrast, it is possible to manufacture complex shapes with 3D printing that are difficult to produce with conventional, i.e., subtractive, methods [21].

Building complete devices with 3D printing is technically possible [22]. However, the reverse engineering of standard parts is not cost- and time-efficient. Thus, many projects integrate 3D printed and off-the-shelf components [23].

### 2.2. Polymers

Polymers that are used in the chemical analysis have to fulfil several conditions. They should be chemically inert against the used media for avoiding interference with the measurements. The used polymers may produce chemical noise, especially at higher temperatures [24]. Further, the 3D printed parts have to resist the physical conditions they are exposed to, such as operating temperature and pressure. For some parts, also the electric properties are critical. 3D printers with dual extruders enable synchronous use of isolating and conductive materials [25].

Below, special technical solutions will be explained in more detail.

**Table 1 metabolites-12-00185-t001:** 3D printed AIMS devices, and their applications. NA—not applicable, ND—not defined. Other abbreviations are listed below.

Device	Polymers	Solvents	Applications	References
**Ion sources and sample separation**				
Cone spray	ESD-safe PETG	Methanol with formic acid	Detection of per- and polyfluoroalkyl substances (PFAS) from soil.	[26,27]
DESI source	PLA	Acetonitrile:water 1:1 (*v*/*v*), methanol:water 9:1 (*v*/*v*), with 0.1% formic acid	Analysis of rat brain tissue and lipid profiles.	[23,28]
DESI support	PLA/PMMA	Acetonitrile:water 1:1 (*v*/*v*)	Analysis of gentamicin sulfate, insulin and chitosan.	[29]
LTP probe	PLA/ABS/PC	NA	*In vivo* monitoring of biosynthesis, direct and multimodal imaging of biological tissues and TLC plates.	[24,30,31]
MasSpec Pen	PDMS	Water	*In vivo* analysis of tissues.	[32]
PSI cartridge	PLA/PP/photopolymer	Solvent mixtures of methanol, water and acetonitrile with 0.1% of formic acid	Analysis of lidocaine and drugs.	[33,34,35,36,37]
PSI cartridge	POM	Methanol:water (1:1)	Direct analysis of complex biological samples.	[38]
PSI cassette	PLA/ABS	Acetonitrile, water, methanol	Monitoring of enzyme reaction for the BuchE detection, two dimensional chromatographic separation for detecting drugs.	[39]
PSI microfluidic device	ABS	Methanol with 0.1% formic acid	Analysis of standard solutions of caffeine, xylose and lysozyme.	[40]
Thread-based electrofluidic device	PMMA	NA	Purification and enrichment of insulin; detection of alkaloids in urine.	[41,42]
**Adapters and holders**				
Chassis of EWOD-based DMF–MS interface	ABS	NA	Chemical reaction monitoring.	[43]
Coupling of DMF to HPLC-MS	NA	Methanol, acetonitrile with acetic acid	On-chip steroid derivatization and automated bioanalyses.	[44]
LTP probe adapter for DESI-MS platform	PLA	NA	Ambient MS imaging of biological samples.	unpublished
PIRL fibre adapter, slice holder, and fibre cleaning channel for a DESI-MS platform	PLA	Water	Dual mode imaging with DESI-MS and PIRL-MS.	[45]
**Ion manipulation and ion mobility spectrometry**				
Drift tube	PLA/PHA/conductive PLA/PETG/ESD-safe PETG	ACN	Detection of tetraalkyl ammonium salts and 2,6-di-tert-butylpyridine.	[25,46]
Electrodes	Conductive carbon nanotube doped polymer	NA	Analysing mixtures of tetraalkyl ammonium bromide salts.	[47]
IMS	PLA/PHA/PETG-CNT/electrically conductive composite PLA	Acetonitrile, Methanol	Detection of tetraalkyl ammonium salts, angiotensin II and bradykinin acetate salts, amphetamines, fentanyls, benzylamines and ketones.	[22,48,49]
Ion funnel	ABS	NA	Proof-of-concept.	[50]
Plastic device for ion separation	PLA/conductive ABS	Acetonitrile, Methanol	Detection and separation of cyclohexylamine, DMPP, tetraalkyl ammonium salts.	[51]
**Robots**				
Open-port probe	PLA	Methanol	Analysis of solid and liquid samples for nebulization gas-based ion sources.	[52]
Purdue Make-It System: Custom plastic plate carriers for DESI-MS platform	ND	NA	High-throughput screening of organic reactions.	[53]
RAMSAY and RAMSAY-2, and sample vials	ABS	Ethanol, acetic acid, water, hydrogen peroxide	Reaction monitoring.	[54,55]
RoSA-MS	ND	NA	Support for robotic surface analysis coupled to an open port sampling interface (OPSI).	[56]
Rotatory multispray holder for nESI	PLA	NA	Reaction monitoring.	[57]

## 3. Ambient Ionisation Sources

### 3.1. Desorption Electrospray Ionisation (DESI)

DESI was the first commercialised AIMS technique. A charged solvent flow or spray is used to desorb and ionise molecules from surfaces. DESI is especially useful for studying relatively large, polar compounds and for imaging [1,58,59,60]. In addition, the mechanism of DESI ion formation and the parameters influencing the detection of compounds has been widely studied [61].

The sample support material affects the reproducibility and sensitivity of DESI analyses. Polytetrafluoroethylene (PTFE) is commonly used; alternative materials such as polymethylmethacrylate (PMMA) [62] and silicon [63] were used for the analysis of plant and animal tissues.

Polylactic acid (PLA) is suitable for creating hydrophobic DESI supports. Elviri et al. [29] analysed insulin, gentamicin sulfate and chitosan from 3D printed PLA supports and compared their performance with PTFE supports. In addition, they tested different sample spot cavities types (cylindrical, cubic, hemispheric). The PLA sample plates provided improved reproducibility, the limit of detection, and linearity. Besides, no memory effect was recorded when washing and reusing the PLA sample supports [29].

In 2020, Zemaitis and Wood [23] reported a 3D printed DESI source consisting of four parts: an angular and a z-distance positioner, a microscope stage (platen), and a structural adapter to the mass spectrometer inlet. They coupled the modular system to a Fourier-transform ion cyclotron resonance (FT-ICR) mass spectrometer. They optimised parameters such as the incidence angle, collection angle, emitter distance from capillary to sample, and the distance of capillary to sample [23]. Subsequently, they used the 3D-DESI source for multimodal mass spectrometry imaging (MSI), analysing phospholipid profiles of rat brain tissues with DESI and matrix-assisted laser desorption/ionisation (MALDI) [28].

A 3D printed, thread-based electrofluidic device for analyte separation and concentration, coupled to an MS analyser with DESI source, separated proteins in 30 min, and insulin was purified from matrix compounds and enriched 10-fold [41]. Furthermore, this low-cost and reproducible analytical platform also could improve the detection of the alkaloids coptisine, berberine and palmatine from urine [42].

### 3.2. Paper-Spray Ionisation
(PSI)

PSI was developed in 2010 by Liu et al. [64]. The sample and solvent are placed on a triangular piece of paper. Applying a direct high-voltage current leads to the formation of a spray on the sharp edge of the paper [64]. Therefore, PSI is an electrospray ionisation method. PSI’s principal advantages are minimal, or no sample pre-processing, low required solvent volume (~10 mL), and short analysis time (10–30 s). In addition, carrier and nebulising gases are not required for PSI. Multiple PSI methods were published in diverse areas like medical, forensic, and food quality control. PSI was used in 20% of 2020’s AIMS papers, highlighting its vast potential for practical applications [65].

Although several commercial platforms are available, PSI is actively investigated by academic groups which build their own prototypes. The critical parameters for PSI systems are the geometry, the tip, the type of paper, and the used solvent system [66]. Off-the-shelf PSI cartridges provide paper support and reservoirs for ensuring a continuous solvent flow [67]. However, the relatively high costs of commercial PSI cartridge solutions suggest 3D printing for creating prototypes.

A 3D printed PSI cartridge from PLA was produced by Salentijn et al. [33]. The 3D-PSI cartridge features a solvent reservoir, a paper tip chamber, a channel for fast wetting, a solvent guide structure, and a cavity. The solvent reservoir provided a continuous solvent flow for several minutes of measurement. The material choice favoured the fast movement of the solvent to the paper tip [33].

Duarte et al. [40] designed a microfluidic device with a 3D microchannel inside a polygon of five sides. A triangular paper tip enables spray ionisation. The base of the microchannel contains a circular solvent reservoir and a high voltage electrode. A special holder helps to focus ions to the mass spectrometer inlet. Using the 3D microfluidic device improved the spray stability, and at least ten minutes of measurement time were possible [40]. Salentijn et al. [33] designed a 3D cartridge for stabilising the aerosol formation. The 3D cartridge has two improvements: sheath gas channels for spray stabilisation and an ion lens guiding the ions to the MS inlet [33].

The ionisation efficiency depends on the type of matrix/paper. Bills et al. [35] used 3D printing to build a spray cartridge which allowed testing different paper types and thin-layer chromatography (TLC) plates. The solvent was supplied through a pipet tip, and a metallic clip applied the high voltage current [35].

Clinical analyses often need an enrichment of the compounds of interest. Zhang and Manicke [68] used a milling machine to create a PSI cartridge with solid-phase extraction (SPE) for sample pre-processing [68]. A re-designed version of this SPE cartridge was fitted to the commercial Prosolia autosampler [36]. Bills and Manicke [37] also mounted a PSI device to a Prosolia autosampler for concentrating and analysing cannabinoids in human body fluids. They built the device from a 3D printed polypropylene and an injection moulded part [37].

3D printed point-of-care devices for early diagnosis are an exciting field for the do-it-yourself (DIY) and medical community. A portable 3D printed enzyme reactor paper-spray (3D ER-PS) cartridge could detect and quantify butyrylcholinesterase (BuChE), a biomarker for metabolic disorders diagnosis in humans serum. The PLA devices could analyse up to six samples. Multiple functions were integrated into this gadget: temperature control, enzyme reaction, analyte transfer, and PSI [39].

Li et al. [69] built a cassette with commercially available electrodes, filter paper, and 3D printed parts. Acrylonitrile butadiene styrene (ABS) and a conductive polymer were used for 3D printing. The device served for analysing drugs from hair and urine. Compounds were separated and enriched by paper chromatography, and the electrodes focussed the ions and improved their transmission to the analyser [69].

PSI can be readily integrated with other techniques, such as microfluidics, solvent and chromatographic separation [70]. 3D printing will have a central role in the development and deployment of innovative PSI systems.

### 3.3. Low-Temperature Plasma (LTP)
Probe

LTP probes use a dielectric discharge barrier to create a plasma beam from a gas flow. LTP can ionise a wide range of low-molecular-weight compounds from solid, liquid, and gaseous samples. Its ionisation capabilities complement electrospray (ESI) and atmospheric pressure chemical ionisation (APCI) [8,71]. In addition, thermal and gas-flow mediated desorption enabled ambient ionisation mass spectrometry imaging (AIMSI) of chemical and biological surfaces with LTP [72,73,74]. Because of the numerous possible applications, LTP has become popular in the research community, and multiple in-house designs exist; for a review, see Martínez-Jarquín and Winkler [75].

In 2016, Martínez-Jarquín et al. [24] published a hybrid 3D printed prototype [24]. Building the 3D-LTP probe with different materials affected its thermal and mechanical stability and the chemical noise in mass spectrometry. PLA, ABS and polycarbonate (PC) were tested and found suitable for chemical analyses and in vivo studies [76]. The 3D-LTP design enables adjusting the tip size and the diameter of the plasma beam, facilitating imaging applications [30,31]. The template is available under a Creative Commons license (https://creativecommons.org, accessed on 27 October 2021) for non-commercial purposes.

## 4. Sampling for Ambient Ionisation Mass Spectrometry
(AIMS)

3D printed gadgets also help in the development of integrated sampling/ambient ionisation methods.

The MasSpec Pen is a handheld probe for the non-invasive extraction of compounds from biological tissues. The tip of the pen was manufactured with 3D printed polydimethylsiloxane (PDMS). At-line analysis of the extracted solutions can assist in decision making during medical surgeries [77].

3D printed cones can be used to collect soil samples, extract and analyse them with electrospray ionisation. This novel approach allows the rapid detection of trace levels of trace per- and polyfluoroalkyl (PFAS) substances and chemical warfare agents (CWAs). Furthermore, building the cones with conductive polymers enables their direct coupling to benchtop and portable mass analysers [26,27].

## 5. Ion Transfer and Ion Mobility Spectrometry
(IMS)

An efficient transfer of ions from the ion source into the mass analyser is critical in AIMS.

A 3D printed flexible ion funnel improved the ion transmission at ambient pressure operation. The device had a similar performance as an ion funnel with conventional construction but reduced manufacturing cost and power consumption [50].

Iyer et al. [47] studied the impact of the geometry on ion transfer, using 3D printed electrodes made from conductive polymers. SIMION (https://simion.com, accessed on 27 October 2021) simulations indicated that higher pressures might be even advantageous for efficient ion transfer (see Figure 2E,F). Although the experimental performance for focusing nanoESI ions, monitored with an ion detection charge couple device (IonCCD) camera, is lower than expected, a high potential of optimising ion transfer tubes with 3D printing is evident [47].

3D printed components are also suitable for the construction of IMS drift tubes.

IMS separates ions travelling in a constant electric field according to their collisional cross-section. The conventional IMS consists of a linear drift tube with ring electrodes and isolators. A high voltage supply generates the electromagnetic field, and a counter-current gas flow can increment the resolution. IMS can employ different ion sources, such as LTP [78] and DART [79]. In addition, the coupling to separation units is possible [80]. No vacuum is required for IMS. Its operation at ambient pressure reduces the requirements for the instrument design and makes IMS an ideal choice for integration into AIMS systems.

3D printing offers new possibilities for studying and producing IMS devices.

A plastic device for ion manipulation with 3D printed electrodes from conductive ABS polymer transmitted after optimisation more than 50% of the spray current to a detector [51].

The resolution of IMS units depends on the length of their drift tube. Schrader et al. [46] simulated and tested 3D printed drift tubes with different combinations of curves. Using drift tubes with chicanes could increase the path length, and therefore improve the resolution of IMS devices whilst permitting a compact instrument design (see Figure 2A–D) [46].

Hollerbach et al. [48] characterised the analytical performance of a 3D printed IMS. PLA and conductive polyethene terephthalate glycol-modified polymer doped with multi-walled carbon nanotubes (PETG-CNT) formed the IMS housing and electrodes. The 3D-IMS demonstrated its functionality in the positive and negative mode for several standard compounds and illicit drugs, such as tetraalkyl ammonium bromide salts (TAA), haloperidol, methamphetamine (MA), 3,4-methylenedioxy-N-ethylamphetamine (MDEA), benzylamines, and sodium alkyl sulfates [48].

The possibility of a mass production of 3D printed IMS devices was demonstrated by the reproducible manufacture of unibody drift tubes consisting of alternating isolating and conductive polymers (PETG and PETG-CNT). Ten IMS drift tubes of a pilot series showed variability of only 0.1% between each other [25].

A double ion gate made the coupling of a 3D printed IMS to any mass analyser possible. Hollerbach et al. [49] demonstrated the suitability of ion trap and quadrupole mass spectrometers for 3D-IMS-MS of amphetamines, opioids, bradykinin and angiotensin II [49].

Drees et al. [22] even built a complete drift-tube IMS with dual extrusion 3D printing. Using non-conductive and conductive PLA polymers enabled the manufacture of all necessary parts: ionisation chamber, Bradbury–Nielsen ion gate, drift tube, and detector [22], underlining the feasibility of building analytical instruments with 3D printing technology.

## 6. Robotics and Imaging

Automation plays a central role in analytical chemistry for increasing sample throughput and improving the reproducibility of methods. Thus, most vendors of analytical instruments also offer autosamplers. Such commercial systems work perfectly fine with a provider’s components and software. However, the integrated platforms are usually costly and difficult to adopt for new applications. On the other hand, 3D printing technology enables the fast and cost-efficient construction of custom robots and controlling them with free software. Furthermore, open licenses allow the copying and modification of these platforms for their projects.

Sampling robots based on 3D printer components are often suitable for ambient ionisation mass spectrometry imaging (AIMSI) since lateral movements in the micrometre scale are possible.

‘RAMSAY’ is an acronym for the ‘robotics-assisted mass spectrometry assay’. The robotic arm is controlled with Arduino (https://www.arduino.cc, accessed on 27 October 2021) and Raspberry Pi (https://www.raspberrypi.org, accessed on 27 October 2021) microcomputers. The RAMSAY delivers vials that are 3D printed from ABS filament. The system can be easily modified and reprogrammed for different purposes. A Venturi pump connected to a metal T-junction was used as an AIMS source [55]. The RAMSAY 2, shown in Figure 3A), provides two robotic arms working synchronously. The system can perform multiple operations and deliver the prepared samples to the MS analyser [54].

A 3D printed robotic arm by Li et al. [56] adapts a 3D line laser scanner to an automated platform (Figure 3B). The Robotic Surface Analysis Mass Spectrometry (RoSA-MS) enables studying the topology of a surface and its chemistry [56].

Mehl et al. [81] built an autosampler for coupling TLC to liquid chromatography (LC) and mass spectrometry. This system that features a 3D printed planar sample holder and open software enabled the activity-directed identification of antibiotics [81].

The MasSpec Pen was redesigned coupled with a da Vinci Xi surgical robot (Figure 3C). The 3D printed tip and case of the MasSpec Pen were optimised for being less invasive and facilitating its adaption for various automated instruments. The device was tested for in vivo tissue analysis during the robotic surgery of a pig [32].

Martínez-Jarquín et al. [14] used a custom-built robot based on RepRap (3D printer) technology for the high-throughput analysis of Mexican Agave spirits (Tequila and Mezcal). Platform movements and analysis time were controlled with G-code. A 3D-LTP ionisation probe enabled measurement times of 10 s per sample [24]. The Open LabBot and its control software RmsiGUI were released as a community development kit [19]. Mounting a 3D-LTP probe and a continuous wave ultraviolet (CW-UV) diode laser for laser desorption (LD) enabled the imaging of alkaloids in plant tissues with a lateral resolution of 50 µm (Figure 4A) [30]. The Open LabBot enables studying relatively large surfaces. Methylxanthines were imaged on TLC plates, covering a total area of 34.8 cm
${}^{2}$
 [31].

## 7. Retrofitting of Existing Platforms

Modifying existing mass spectrometry systems with 3D printed parts can augment their functions with a very modest financial investment. Some examples of upgrading commercial platforms with 3D printed AIMS sources and robotics were already given in Section 3 and Section 6.

Pulliam et al. [57] developed a 3D printed rotatory multi-spray holder for sampling and analysing multiple reaction vessels. The 3D printed holder was manufactured from PLA. An Arduino Uno controlled the rotatory stepper motor. For analysis, they coupled a Mini 12 miniature mass spectrometer [38]. The system allowed the synchronous monitoring of up to six reactions with high reproducibility and without carryover [57].

Fitting a 3D printed open-port probe (OPP) to a commercial PAL-HTC-xt autosampler (CTC Analytics) increased the sample throughput and method robustness for analysing liquid and solid samples [52]. In addition, the OPP was printed with stereolithography (SLA) and a methanol-resistant resin. As a result, the platform directly detected pesticides from fruit peel surfaces, illegal pills, and urine and plasma compounds [52].

The Purdue Make-It System is a modified Prosolia DESI 2D imaging stage for the high-throughput screening of organic reactions. Several parts were manufactured with 3D printing. The system is capable of analysing a plate with 384 reaction mixtures in 7 min [53].

Digital microfluidics (DMF) enables de handling of picoliter-scale droplets. Hu et al. [43] coupled an ‘electrowetting on dielectric’ (EWOD) DMF device to a mass spectrometer for monitoring the oxidation of glutathione by hydrogen peroxide: 2 GSH + H
${}_{2}$
O
${}_{2}$
-> GSSG + 2H
${}_{2}$
O. For mounting the microchip to the Venturi easy ambient sonic-spray ionisation mass spectrometry (V-EASI-MS), they used a 3D printed adaptor [43].

A 3D printed manifold was also used to couple DMF with high-performance liquid chromatography (HPLC)-MS. The platform is suitable for pretreating samples, for example, reaction mixtures in aqueous buffers and tryptic digests [44].

Multimodal ionisation increases the range of detectable compounds and offers additional analytical possibilities. For example, limonene and its ozone-initiated reaction products can be studied with LTP-MS [82]. Adopting a 3D-LTP probe to a Prosolia DESI 2D imaging stage with a 3D printed adaptor(Figure 4B) resulted in a system that is suitable for imaging limonene and related compounds from a lemon peel (ongoing work).

Katz et al. [45] built a multimodal mass spectrometry imaging (MSI) platform that, in addition to the standard DESI source, provides sampling with a picosecond infrared laser (PIRL). First, 3D printed parts were used to mount the PIRL fibre on a Waters imaging stage (Figure 4C). Then, using the provided hardware and software, complementary molecular images of cancer tissue with a spatial resolution of 200 × 200 µm
${}^{2}$
 were obtained [45].

## 8. Sustainability of 3D
Printing

Assessing the sustainability of 3D printing is complex. Compared to conventional subtractive manufacturing, fused deposition modeling is more efficient in the use of materials, and less waste is generated. Balancing the energy consumption of different manufacture processes, such as injection molding, machining and 3D printing is more complicated, because one has to take into account the complete production chain, which includes also storage and transport [83]. Developing polymers for 3D printing which are biocompatible, recyclable, and degradable, and the reprocessing of plastic waste will be pivotal for making 3D printing ‘green’ [84], M. Maines et al. [85], Zhao et al. [86].

## 9. Current State and
Perspectives

3D printing enables the fast assembly of custom ambient ionisation mass spectrometry (AIMS) systems. The analytical performance of some set-ups is already comparable with commercial systems and suitable for routine use. The increased availability of 3D printers, their finer lateral resolution and new materials will enable more complex devices, for example, to improve the ion transfer in AIMS.

Using 3D printing drastically shortens development life cycles, and sharing of 3D printing files speeds up the implementation of novel devices in research labs around the globe without the need of first establishing industrial manufacturing and commercial distribution.

Upgrading existing mass analysers with 3D printed ambient ionisation sources enables multimodal experiments with low economic effort. Besides, 3D printing components and software can be used to build sampling and imaging robots. However, although integrating custom AIMS prototypes into existing commercial platforms is highly attractive, it could be troubled by proprietary hardware and software standards. Commercial providers of mass analysers could support the advance of AIMS techniques by offering developer-friendly interfaces and adhering to community standards such as HUPO file formats (https://www.psidev.info, accessed on 27 October 2021).

Especially the development of polymers which are optimized for analytical instruments, and the printing with materials such as metals, PEEK, glass and ceramics could further boost the role of 3D printing in analytical chemistry and improve its sustainability.

## Figures and Tables

**Figure 1 metabolites-12-00185-f001:**
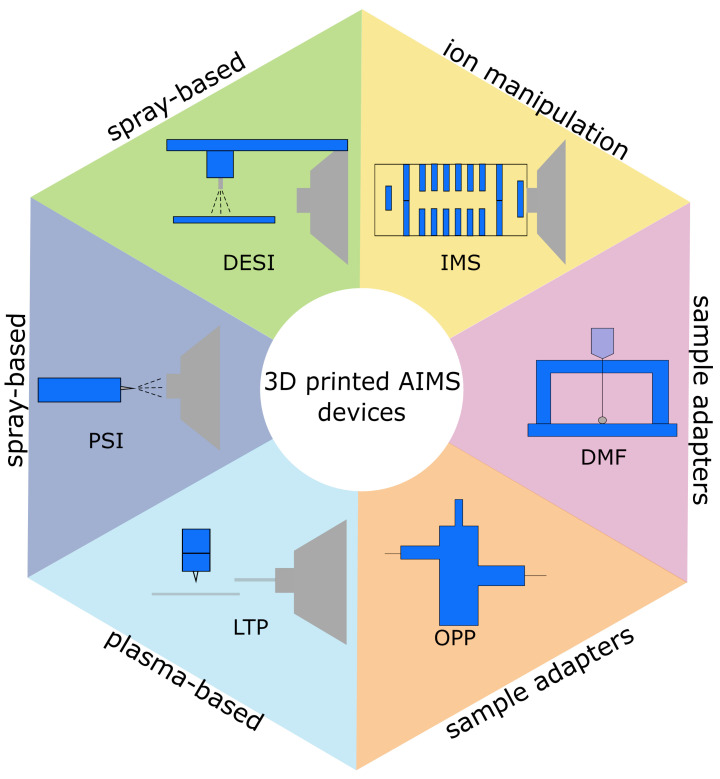
3D printed components (indicated in blue) for ambient ionisation mass spectrometry (AIMS). Desorption electrospray ionisation (DESI) source; digital microfluidics (DMF) chip for automation of sample preparation; 3D printed drift tube for ion mobility spectrometry (IMS); 3D printed low-temperature plasma (3D-LTP) probe; open port probe (OPP) for sampling with spray-based ion source; paper-spray ionisation (PSI) cartridge, supporting the paper tip and facilitating the solvent application.

**Figure 2 metabolites-12-00185-f002:**
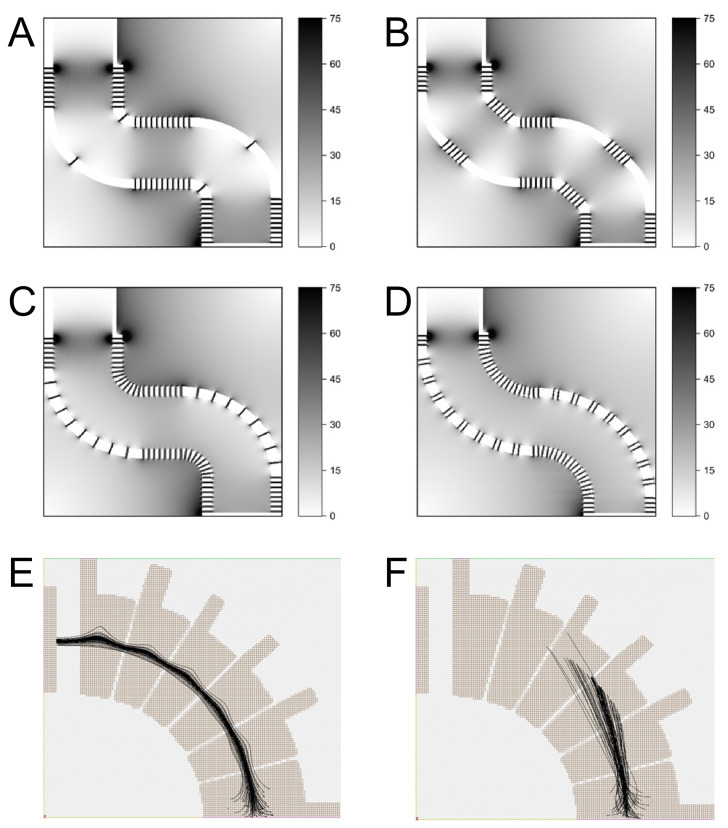
Simulation of the electric field strength (V/mm) (**A**–**D**) and spatial ion traveling (**E**,**F**) in 3D printed curved electrodes. (**A**) Two turns with two consecutive 45° electrodes; (**B**) four turns with separated 45° electrodes; (**C**) two turns with nine consecutive 10° electrodes; (**D**) 18 separated 10° electrodes. Adapted from [46], with the permission of Elsevier, copyright 2020. Curved ion focussing electrodes (**E**) with atmospheric collision gas; (**F**) in vaccum. Without collision, ion transmission is prevented. Reprinted from [47], with permission from the American Chemical Society, copyright 2019.

**Figure 3 metabolites-12-00185-f003:**
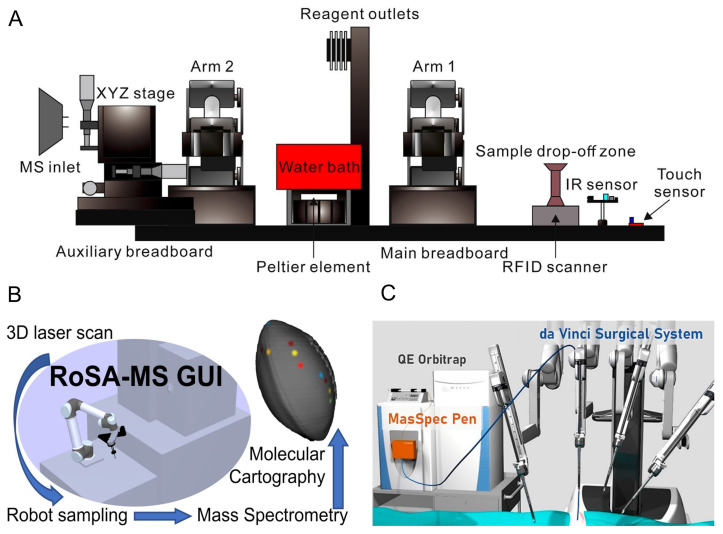
Robotic systems with 3D printed components and open-source software. (**A**) Dual robotic arms in front of an ion trap. The arms prepare and deliver the samples. Automation of multiple sample preparation steps with touch and infrared (IR) sensors, water bath and a XYZ stage. Reprinted from [54], with permission of Elsevier. (**B**) The Robotic Surface Analysis Mass Spectrometry (RoSA-MS), coupled with a 3D surface sampling enables surface contour digitalization and 3D molecular cartography. The RoSA-MS has a modular design allowing modifications for diverse applications. Reprinted from [56], with the permission of the American Chemical Society, copyright 2018. (**C**) Implementation of the laparoscopic version of the MasSpec Pen, coupled to the da Vinci X Surgical system for in vivo tissue analyses. Reprinted from [32], with the permission of the American Chemical Society, copyright 2020.

**Figure 4 metabolites-12-00185-f004:**
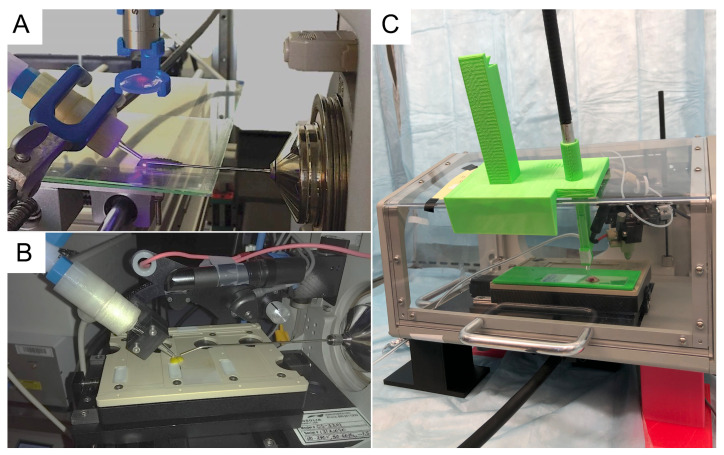
Imaging platforms with 3D printed components. (**A**) LD-LTP MS imaging setup. The system uses a 3D printed holder for the laser and lens, a 3D-LTP probe, and an Open LabBot movement platform. Reprinted from [30], with the permission of the American Chemical Society, copyright 2019. (**B**) Mounting of a 3D-LTP probe on a Prosolia DESI platform for imaging terpenes, using a 3D printed adapters. (**C**) Adaptation of a commercial Waters DESI-MS for dual-mode (DESI and laser desorption) imaging. Parts in green and the legs for supporting the DESI-MSI platform inside a safety cabin were 3D printed. Reprinted from [45], with the permission of the American Chemical Society, copyright 2020.

## Data Availability

Not applicable.

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
