# Peer review of "Build, Share and Remix: 3D Printing for Speeding Up the Innovation Cycles in Ambient Ionisation Mass Spectrometry (AIMS)"

_metabolites, 2022, doi:10.3390/metabo12020185_

Round 1

Reviewer 1 Report

The proposed manuscript gives a good account of the utility of 3d printing technologies in ambient mass spectrometry and clearly delineates the wide spread of applicability and the powerful customization offered by 3d printing.

It is rather similar to the recent minireview by some of the same authors in Analytical Methods (DOI:10.1039/D0AY02290J) and is a good complement to the review by Salentijn et al. (https://doi.org/10.1016/j.aca.2021.338332).

I would suggest to the authors to maybe highlight a bit more the impact of these technologies in the specific field of metabolites research, in an effort to differentiate the current manuscript from the 2 mentioned above.

In addition there may be other ‘educational applications’ that may be worth discussing with respect to particle traps.

https://skoltech.mit.edu/collaborative-projects/next-generation-program/first-round-ngp-grants/3d-printed-miniaturized

https://scoollab.web.cern.ch/3d-printable-quadrupole-ion-trap

Otherwise, I find this manuscript well suited for publication in Metabolites with the minor revision suggested.

I only noted a couple minor typos listed below:

  • Line 154: word ‘found’ my be missing in “..were tested and suitable..”
  • Line296: may I suggest “…DESI source, provides sampling with a picosecond infrared laser…”?

Author Response

Referee 1

=========

The proposed manuscript gives a good account of the utility of 3d printing technologies in ambient mass spectrometry and clearly delineates the wide spread of applicability and the powerful customization offered by 3d printing.

It is rather similar to the recent minireview by some of the same authors in Analytical Methods (DOI:10.1039/D0AY02290J) and is a good complement to the review by Salentijn et al. (https://doi.org/10.1016/j.aca.2021.338332).

I would suggest to the authors to maybe highlight a bit more the impact of these technologies in the specific field of metabolites research, in an effort to differentiate the current manuscript from the 2 mentioned above.

Authors

=========

Thank you! We were not aware of the recent article of Salentijn et al. and included it in our paper. Indeed, our perspective is complementary to existing literature and should highlight the potential of 3D printing in Ambient Ionisation MS.

We partly rearranged the introduction and first section. In addition, we added references, why AIMS methods are attractive for metabolite research: the detection of a more extensive range of compounds (e.g. of high hydrophobicity, volatiles and semi-volatiles) without the need of the work-up of native biologica samples.

Referee 1

=========

In addition there may be other ‘educational applications’ that may be worth discussing with respect to particle traps.

https://skoltech.mit.edu/collaborative-projects/next-generation-program/first-round-ngp-grants/3d-printed-miniaturized

https://scoollab.web.cern.ch/3d-printable-quadrupole-ion-trap

Authors

=========

We greatly sympathise with these educational projects, but our scope is more directed towards 'professional' use of 3D printing in research and industries and, in this particular manuscript, its use in AIMS. But hopefully, one day, we can print our MS analysers!

Referee 1

=========

Otherwise, I find this manuscript well suited for publication in Metabolites with the minor revision suggested.

Authors

=========

Thank you very much for your positive judgment!

Referee 1

=========

I only noted a couple minor typos listed below:

Line 154: word ‘found’ my be missing in “..were tested and suitable..”

Authors

=========

Thanks! We inserted ‘found’ before ‘suitable’.

Line296: may I suggest “…DESI source, provides sampling with a picosecond infrared laser…”?

Authors

=========

Good point! We inserted ‘sampling with’ before ‘a picosecond’.

Reviewer 2 Report

In general, the manuscript is well-written and offers an interesting perspective. I believe that while it perhaps does not exactly match the topical outline of Metabolites, it could nonetheless be of interest to its readers. 

I would ask the Authors to consider the following remarks:

- Introduction, first two paragraphs: the Authors juxtapose "conventional" MS techniques, which almost always involve the use of sample preparation techniques but allow high selectivity, with AIMS techniques which, while limited in many ways, do have an advantage in that they allow direct analysis of biological samples. However, they fail to mention other direct MS techniques which in many ways outperform AIMS in the discussed application, such as SIFT-MS and, perhaps more importantly, PTR-MS. This could give the reader a false sense that AIMS techniques are the go-to alternative to established MS methods for direct analysis of metabolites.

-line 13: "high sensibility and selectivity" - I think (in the context of MS) the Authors mean sensitivity, not sensibility.

- line 45: the issue with overhangs is true for FDM or SLA printers, but not so much for e.g. powder bed fusion printers.

- line 183: this description of the principle of operation of IMS is not accurate. The crucial parameter in IMS is not the size of the molecule itself, but rather its collisional cross-section. Conversely, I'm not sure if IMS can be considered an MS technique.

- The fact that most 3D printing materials, particularly those for FDM and SLA printing, are not chemically inert was hinted at in the manuscript, but this issue was not expanded upon. However, thermoplastic polymers such as PLA or ABS emit a significant amount of different VOCs (such as BTEX compounds in the case of the latter) even at temperatures just above 100°C, and even at ambient temperatures. This could introduce substantial background noise in MS analysis and adversely affect the LOD in metabolomic studies. There are now commercially available FDM printers with high-temperature extruders which allow printing with more inert materials such as e.g. PEEK, and also techniques that allow additive manufacturing using metal. I think that this issue should be discussed in the manuscript. 

Author Response

Referee 2

=========

In general, the manuscript is well-written and offers an interesting perspective. I believe that while it perhaps does not exactly match the topical outline of Metabolites, it could nonetheless be of interest to its readers.

I would ask the Authors to consider the following remarks:

Referee 2

=========

- Introduction, first two paragraphs: the Authors juxtapose "conventional" MS techniques, which almost always involve the use of sample preparation techniques but allow high selectivity, with AIMS techniques which, while limited in many ways, do have an advantage in that they allow direct analysis of biological samples. However, they fail to mention other direct MS techniques which in many ways outperform AIMS in the discussed application, such as SIFT-MS and, perhaps more importantly, PTR-MS. This could give the reader a false sense that AIMS techniques are the go-to alternative to established MS methods for direct analysis of metabolites.

Authors

=========

Yes, we agree that there are excellent methods for the direct detection of volatile organic compounds. Therefore, we added a paragraph into the introduction to clarify that we refer to AIMS methods for analysing complex and non-volatile metabolites.

Referee 2

=========

-line 13: "high sensibility and selectivity" - I think (in the context of MS) the Authors mean sensitivity, not sensibility.

Authors

=========

Of course! Thanks; We changed ‘sensibility’ to ‘sensitivity’.

Referee 2

=========

- line 45: the issue with overhangs is true for FDM or SLA printers, but not so much for e.g. powder bed fusion printers.

Authors

=========

True! We added a phrase clarifying that.

Referee 2

=========

- line 183: this description of the principle of operation of IMS is not accurate. The crucial parameter in IMS is not the size of the molecule itself, but rather its collisional cross-section. Conversely, I'm not sure if IMS can be considered an MS technique.

Authors

=========

Sorry for this sloppy IMS definition. We corrected that. IMS is not an MS method but combines well with ambient ionisation and adds an analytical separation to MS analysers. Furthermore, the differences between ion transport/ manipulation/ IMS are sometimes tiny. Thus, we think some information on 3D printed IMS parts is helpful for the readers interested in metabolite analyses/ AIMS.

Referee 2

=========

- The fact that most 3D printing materials, particularly those for FDM and SLA printing, are not chemically inert was hinted at in the manuscript, but this issue was not expanded upon. However, thermoplastic polymers such as PLA or ABS emit a significant amount of different VOCs (such as BTEX compounds in the case of the latter) even at temperatures just above 100°C, and even at ambient temperatures. This could introduce substantial background noise in MS analysis and adversely affect the LOD in metabolomic studies. There are now commercially available FDM printers with high-temperature extruders which allow printing with more inert materials such as e.g. PEEK, and also techniques that allow additive manufacturing using metal. I think that this issue should be discussed in the manuscript.

Authors

=========

Indeed! We integrated additional comments in the general section (overview) and at the end of the conclusion (as well, some comments related to sustainability).

Reviewer 3 Report

I carefully read the article entitled "Build, share and remix: 3D printing for speeding up the innovation cycles in ambient ionisation mass spectrometry (AIMS)".

I have found this review very interesting, presenting a very important topic, in a field that is gaining increasing interest and it can be considered an ongoing great project.

The review is very well documented and the original articles are presented in a brief but intriguing way. My only point is that in my opinion summarizing tables containing the different polymer used, printing technique, solvent supply -yes/no and other key paryameters should be added at the end of paragraph 2.1, 2.2 and 4.

However, I found the article very interesting and well-written so I strongly recommand its publication in Metabolites

Author Response

Referee 3

=========

I carefully read the article entitled "Build, share and remix: 3D printing for speeding up the innovation cycles in ambient ionisation mass spectrometry (AIMS)".

I have found this review very interesting, presenting a very important topic, in a field that is gaining increasing interest and it can be considered an ongoing great project.

The review is very well documented and the original articles are presented in a brief but intriguing way. My only point is that in my opinion summarizing tables containing the different polymer used, printing technique, solvent supply -yes/no and other key paryameters should be added at the end of paragraph 2.1, 2.2 and 4.

However, I found the article very interesting and well-written so I strongly recommand its publication in Metabolites

Authors

=========

Thank you very much for your positive feedback. Based on your comment, we added a table which summarises 3D printed devices for AIMS, used solvents and applications.

Reviewer 4 Report

The perspective, "Build, share and remix: 3D printing for speeding up the innovation cycles in ambient ionisation mass spectrometry (AIMS)" by Garcia-Rojas et al. is a well researched review about 3D printing within AIMS that will be of interest to multiple readers. That said there are issues that need to be addressed prior to further consideration and I do not recommend publication unless the perspective is rewritten. 

First and foremost, the perspective reads like a laundry list of scientific manuscripts, which is problematic for a perspective. Perspectives should include ssome review elements but should have substantial input/criticism on the literature and overall direction of the field. Many of the paragraphs begin with some other et al. did some achievment, but doesn't go farther than that. It's also a little repeative since most of the manuscript is written in that format. 

Other comments:

Page 1 18 Specify differences

Page 2 38-39 Incomplete sentences. Also it may be worth noting that there are non-hybrid constructions as well, so it may be worth just saying "constructions".

67-68 authors list that certain materials are favored for animals and plants but don't explain why. It is confusing given they go on to say that PLA  has many qualities that make it useful. 

Figure 1

The authors include ion mobility, but this isn't really a source, which the center of the figrue inidicates.

Page 4 Line 93-95  "principal advantages are minimal, or no sample, preprocessing" is poorly written and cannot be interpreted correctly. As stated, it is saying PSI's principal advantages are minimal, except for all the things the authors list.

Page 5 Line 182 seems to be a random sentence. There are several like this that are confusing. 

Author Response

Referee 4

=========

The perspective, "Build, share and remix: 3D printing for speeding up the innovation cycles in ambient ionisation mass spectrometry (AIMS)" by Garcia-Rojas et al. is a well researched review about 3D printing within AIMS that will be of interest to multiple readers. That said there are issues that need to be addressed prior to further consideration and I do not recommend publication unless the perspective is rewritten.

Authors

=========

Thank you for your positive and constructive feedback.

Referee 4

=========

First and foremost, the perspective reads like a laundry list of scientific manuscripts, which is problematic for a perspective. Perspectives should include ssome review elements but should have substantial input/criticism on the literature and overall direction of the field. Many of the paragraphs begin with some other et al. did some achievment, but doesn't go farther than that. It's also a little repeative since most of the manuscript is written in that format.

Authors

=========

We expanded the manuscript with new aspects, such as the sustainability of 3D printing and speculated on future directions of the field in the final part. Further, we rearranged and rewrote several parts to make the text more appealing to read.

Referee 4

=========

Other comments:

Page 1 18 Specify differences

Authors

=========

We added a paragraph about the different desorption/ionisation mechanisms of spray-based, plasma-based and laser/post-ionisation methods.

Referee 4

=========

Page 2 38-39 Incomplete sentences. Also it may be worth noting that there are non-hybrid constructions as well, so it may be worth just saying "constructions".

Authors

=========

We reorganized this part. Now, we only have two sections left: ‘Part design’ and ‘Polymers’.

Referee 4

=========

67-68 authors list that certain materials are favored for animals and plants but don't explain why. It is confusing given they go on to say that PLA has many qualities that make it useful.

Authors

=========

We revised this section. Now, we leave it open if, e.g. silicon is better for animal and plant samples since comprehensive data are still unavailable.

Referee 4

=========

Figure 1

The authors include ion mobility, but this isn't really a source, which the center of the figrue inidicates.

Authors

=========

OK. We changed 'AIMS sources' to 'AIMS devices'.

Referee 4

=========

Page 4 Line 93-95 "principal advantages are minimal, or no sample, preprocessing" is poorly written and cannot be interpreted correctly. As stated, it is saying PSI's principal advantages are minimal, except for all the things the authors list.

Authors

=========

Thanks! We changed the text to ‘principal advantages are no, or minimal, sample pre-processing’. This should be clear now.

Referee 4

=========

Page 5 Line 182 seems to be a random sentence. There are several like this that are confusing.

Authors

=========

This sentence starts with a new topic. The paragraph before explains the possibilities to use 3D parts for ion transfer/manipulation.

The mentioned sentence states that also IMS devices can be built. To make this more clear, we removed the line break. The sentence is now connected with the rest of the topic of IMS.

Reviewer 5 Report

The paper is extremely interesting and fulfill with the Journal aim and scope. The text is clear and complete and all aspects were properly reported.

I have just some minor remarks before the acceptance of the paper, as below reported:

  • please insert also tables with applications (biological, environmental, etc..)
  • highlight better the advantage in term of green chemistry

Author Response

Referee 5

=========

The paper is extremely interesting and fulfill with the Journal aim and scope. The text is clear and complete and all aspects were properly reported.

I have just some minor remarks before the acceptance of the paper, as below reported:

please insert also tables with applications (biological, environmental, etc..)

Authors

=========

Thanks for your positive and constructive review. We added a table, which lists reported 3D printed devices for AIMS and applications.

Referee 5

=========

highlight better the advantage in term of green chemistry

Authors

=========

We added a section about the sustainability of 3D printing. However, evaluating the environmental impact of 3D printing is complex, and there certainly are research opportunities, e.g. in the development of eco-friendly materials.

Round 2

Reviewer 4 Report

The perspective has been editted and reads much better now. I think the additions the authors made enhance the manuscript a lot. That said, I will push back on the "perspective" aspect of the manuscript. 

I still do not find that the mansucript is a perspective. It really is a review as the level of critism and perspective provided by the authors are insufficient. The most I can find is located in the conclusions.

That said, there is nothing scientifically wrong in the review, so I suggest it being reclassifed but will default to the aditor's opinion.